# Flow Mechanism Characterization of Porous Oil-Containing Material Base on Micro-Scale Pore Modeling

**DOI:** 10.3390/ma14143896

**Published:** 2021-07-13

**Authors:** Ke Yan, Tingting Yin, Jiannan Sun, Jun Hong, Yongsheng Zhu

**Affiliations:** Key Laboratory of Education Ministry for Modem Design and Rotor-Bearing System, Xi’an Jiaotong University, Xi’an 310027, China; ytt183@126.com (T.Y.); sun392802414@stu.xjtu.edu.cn (J.S.); jhong@mail.xjtu.edu.cn (J.H.); yszhu@mail.xjtu.edu.cn (Y.Z.)

**Keywords:** micro-scale pores modeling, porous oil-containing cage, flow mechanism characterization, rand seeds theory, micro-scale resistance model

## Abstract

The self-lubricating effect of the porous oil-containing cage is realized by storing and releasing lubricants through its internal micro-scale pore structure. The internal flow and heat transfer process in the micron-submicron pore structure is crucial to the self-lubricating mechanism of the porous oil-containing cage. To this end, a new modeling method of porous cage was proposed based on random seeds theory, and the local two-dimensional models of porous cage with different micro-scale pore structure were established. The multiphysics coupling simulation analysis of lubricating oil inside the porous cage with the effect of centrifugal force and thermal expansion was carried out based on the COMSOL Multiphysics platform. In order to characterize the micro-scale pore structure, new structural parameter indicators, such as relative surface perimeter, effective porosity, tortuosity and fluid properties related to the internal flow process, were all extracted from the above models. Combing with the Hagen–Poiseuille equation, a flow resistance model of oil flow inside the porous oil-containing cage was obtained. Finally, comparison of simulation results and analytical solutions of the micro-scale resistance model was carried out to verify the correctness of the micro-scale resistance model. The work provides a new direction for the study of the lubrication mechanism of the porous oil-containing cage.

## 1. Introduction

Because of its excellent self-lubricating properties, the porous oil-containing cage is widely used in support bearings of major equipment in field of aerospace and precision slewing. During bearing service period, lubricants which are stored in the micron-submicron pores of the porous cage flow out under the effect of the centrifugal force and thermal effect. When bearing speed slows down or the temperature drops, the capillary force of micro-scale pore structure plays a key role to suck back the lubricating oil, thus, the self-lubrication of the porous oil-containing cage is realized [1]. As the speed of the cage rises, the lubricant overflows from the micro-pores of the cage under the combined effect of centrifugal force and thermal effect. Then, when the speed of the cage decreases or the cage stops working, the lubricant is sucked back into the cage and stored under the capillary action of the micropores, thus realizing the circulating lubrication of the lubricant in the porous oil-containing cage. It can be seen that the shape of the micro-scale pore structure, the material properties of the lubricating medium and the coupling effect of the flow and heat transfer directly affect the oil flow resistance characteristics inside the porous structure, which is key to maintaining a good self-lubricating performance for the porous cage.

At present, the research literature on the lubrication performance of porous oil-containing cages mainly focuses on experimental studies. For example, Yan [2] and Shi [3] investigated the friction and wear characteristics of porous cage with different pore-forming agent content. Results show that the content of pore-forming agents is proportional to the size of the micro-scale pore structure, which can be used to control the lubrication performance of the porous oil-containing cage. Through scanning electron microscope images, Qiu [4] found that the porous PI material presents a through-hole structure with an “ink bottle” shape, and it has a uniform pore size distribution. The experimental work shows that the value of porosity directly determines the oil content performance of the oil-containing material. From the above experimental studies, it can be concluded that porous structure has a great influence on the lubrication performance of the porous oil-containing material. However, experimental research mainly focuses on the effects of different oil content and porosity on its tribology performance and self-lubrication from a macro perspective. However, analysis of the oil flow process and mechanism inside the micro-scale pores, especially with the effect of multi-force field, is still lacking. This is due to the structural characteristics of porous materials, such as randomness, complexity and micron/submicron scales, thus leading to great difficulties in structural modeling. To this end, a porous model was reconstructed by X-ray CT scanning technology for the porous material [5,6]. Based on the reconstruction data of the porous model, the geometry and interconnection data of the porous model were extracted. For example, Celik et al. found the volume-averaged (i.e., macroscopic) transport properties, such as permeability and inertia coefficient, of two aluminum foams with 10 and 20 pores per inch (PPI) pore density using microtomography images and further adopting a variety of inspection methods to verify the results obtained. [7] Furthermore, a simplified geometric model of porous materials was established by the authors of [8,9] by using the maximum sphere method to turn the microchannel into pores and throats. Since the porous oil-containing cage is generally formed by cold-press sintering, its internal pore size is between 0.5 μm and 5 μm [10]. However, the highest accuracy of the current X-ray technology is 0.699 μm, which makes it difficult to precisely scan and identify the micro-scale pore structures, causing the reconstruction model seriously distorted.

In the analysis of the internal flow mechanism of porous materials, the commercial software is usually employed to simulate the flow process in porous media. For instance, Wang [11] theoretically analyzed the seepage equation and heat conduction conditions in porous medium, and the simulation analysis were carried out by ANSYS, the seepage process and stress field in porous medium under special circumstances were obtained. Based on COMSOL Multiphysics platform, Yue [12] analyzed the seepage law in porous media, and the result showed that the seepage of lubricating oil in the material was mainly related to the pore structure and external energy. Nevertheless, due to the particularity of the porous cage structure, the current micro-scale pore models of flow analysis on the porous material are not accurate, and the detailed characterization of the porous structure is lacking.

In order to investigate the seepage mechanism in porous media and solve the continuity equation, momentum equation and energy equation of fluid flow, the porous media were often regarded as continuous media and their micro-structure characteristics were ignored. The common theoretical models for the seepage law of porous media include Darcy’s law, the Nonlinear Permeability law and the Ergun equation (K1 = 150, K2 = 1.75) [13,14,15], which were all derived from a number of particle bed experimental studies. Therefore, these models were all empirical and semi-empirical equations to describe the permeation laws of porous media. According to those equations, researchers proposed different empirical coefficients. For example, in MacDonald’s [16] study, K1 = 180 and K2 = 1.8. Wyllie [17] recommended K1 = 172.8 by calculating the characteristic length and shape factor of the particles. Furthermore, Fand [18] studied the parameters K1 and K2 of the Ergun equation and found that when the Reynolds number range was 5 to 80, K1 = 182 and K2 = 1.92, while the Reynolds number was greater than 120, K1 = 225 and K2 = 1.61. From these studies, it can be concluded that with different research object and experiment conditions, the empirical equations for flow resistance loss in porous media have different empirical coefficients, indicating that the current empirical and semi-empirical equations have large errors and fail to accurately represent the permeability law in general porous media.

Based on the above discussion, the traditional flow resistance model and related theories cannot be directly applied to the study of the seepage mechanism in porous medium. In this paper, a new modeling method of porous cage was proposed based on random seeds theory, and the local two-dimensional models of porous cages with different micro-scale pore structure were established. In order to characterize the micro-scale pore structure, new structural parameters, such as relative surface perimeter, effective porosity, tortuosity and fluid properties related to the internal flow process, were all extracted from the new models. The verification between the flow resistance model inside porous medium and the simulation results shows the effectiveness of the method proposed in this paper.

## 2. Micro-Scale Pores Modeling

### 2.1. Modeling Theory of Random Seeds

For the oil-containing cage, the size and distribution of pores in porous material depend on the ratio of the raw materials and preparation process. During the sintering and pressing process, the raw material powder is deformed and bonded by the influence of the pressing pressure, sintering temperature, and holding time, leading the porous structure to be formed by the pores between the particles. Therefore, the shape of the raw material particles is no longer circular from the perspective of the cross section. In order to improve the reduction degree of the two-dimensional cross-sectional model, random seeds are set to elliptical shape. Based on the preparation process of porous oil-containing cages and the principle of pseudo-random number generator, a random seed modeling method for establishing a two-dimensional cross-sectional model of porous media is proposed as follows.
According to the proportion (*P%*) and the size (*d_g_*) of the raw material particles of the porous oil-containing cage, the size (*a_s_, b_s_*) and the number (*N_s_)* of seeds is set as *a_s_ × b_s_ ≈ d_g_ × d_g_*, *N_s1_:N_s2_:... = p_1_%:p_2_%:…*;The size of modeling space (the side length was set to *L*), which is the space for dispersing the seeds *(L^2^)*, is set and the seed placement position *(x_m_, y_m_)* as in Equation (1) is defined;
(1)$$\left(x_{m},y_{m}\right)\in L^{2}$$The random function *(math.rand())* is used to generate the random position coordinates of the seeds as $\left\{\left(x_{1},y_{1}\right),\left(x_{2},y_{2}\right),\ldots ,\left.\left(x_{m},y_{m}\right)\right\}\right.$, and the oval seeds of specified size *(a_s_, b_s_)* or random size *(a_rand_, b_rand_)* at the position points are also generated;The oval seeds overlap situation has been checked by Equation (2). If oval seeds intersect, it will be processed by the union operation of Boolean operations *(*$seed_{A}\cup seed_{B}$*)*, which merges the multiple entities into one entity to imitate the extrusion process of the raw material particles;
(2)$$\left\{\begin{array}{c} \left|x_{m}-x_{m-1}\right|<min\left(0.5d_{sm},0.5d_{s\left(m-1\right)}\right)\\ \left|y_{m}-y_{m-1}\right|<min\left(0.5d_{sm},0.5d_{s\left(m-1\right)}\right) \end{array}\right.$$If the oval seeds have overflown in the modeling space, as in Formula (3), Boolean operation of deleting the overflow part $\left(seed_{A}-C_{U}\left(L^{2}\right)\right)$ will be performed;
(3)$$\left\{\begin{array}{c} x_{m}+0.5d_{sm}>L,\;x_{m}-0.5d_{sm}<0\\ y_{m}+0.5d_{sm}>L,\;y_{m}-0.5d_{sm}<0 \end{array}\right.$$

Step 4 and Step 5 are repeated until all the overlapping oval seeds are merged into one entity and the oval seeds beyond the modeling space are cut off and deleted. Finally, the modeling space (*L^2^*) is used to subtract the seed accumulation model to obtain the pore between the seeds, which is the final micro-scale pore model.

### 2.2. Modeling Process of Random Seeds

Based on the model theory of random seeds, the platform of COMSOL Multiphysics is used to carry out the modeling research of the two-dimensional cross-section with porous materials. The modeling research is completed by using the *Math.random* function of the Java^®^ Language Specification and the algorithm flowchart is shown in Figure 1.

First, the seed number *N_s_* is set and the modeling space *L^2^* (*L* = 100 μm in this paper) is specified. In the literature [19], the three-dimensional internal structure of the porous cage has been scanned and observed using X-ray CT technology, and it has been concluded that the small pores within the porous cage account for the majority of the pores, and the dense arrangement of these small pores is the main reason for the connectivity of the pores in porous oil containing cage. Therefore, the size of seeds is divided into two categories, including small-size seeds (the number was *N_s1_*) and large-size seeds (the number was *N_s2_*) which satisfies *N_s1_ s> N_s2_* based on the proportion (*P%*) of raw material particles.

Because the return value of the *Math. random()* function is a random double-precision floating point number whose value is between 0.0 and 1.0, the size of the seed generated with this function will be in a fixed interval satisfying $d_{rand}\in \left[min,max\right]$. At the same time, the random position coordinates $\left\{\left(x_{1},y_{1}\right),\left(x_{2},y_{2}\right),\ldots ,\left.\left(x_{m},y_{m}\right)\right\}\right.$ of seeds are generated in the modeling space *L^2^* by the *Math.random()* function. The random size of oval seeds is combined with the random position coordinates which further replicates the true formation process of pore structure in porous materials.

According to Equations (2) and (3) of random seeds modeling theory, the distribution of random seeds in the modeling space is checked, and Boolean operations are used to unite the overlapping graphics ($seed_{A}\cup seed_{B}$) and delete the beyond graphics $\left(seed_{A}-C_{U}\left(L^{2}\right)\right)$. Figure 2 shows the formation process of porous model with an effective porosity of 0.1913 as an example, where (a) displays the accumulation drawing of oval seeds in the modeling space, and (b) is the forming drawing in which exhibits the subtraction of the accumulation of oval seeds from the modeling space.

Figure 3 shows a binary graph of porous material, which is extracted from the scanning electron microscopy image of porous oil-containing cage. By comparing Figure 3 and Figure 2b, the morphological characteristics and distribution of the two figures are essentially the same, which shows the rationality of using the random seeds modeling theory to establish a two-dimensional local model of porous oil-containing cages.

## 3. Model Simulation

Taking the porous oil-containing cage (polyimide material) of 7008C bearing as the research object, based on the two-dimensional local model of porous oil-containing cages, the flow law of lubricating oil in the porous oil-containing cage, which is under the influence of centrifugal force and thermal effect, is simulated by the COMSOL Multiphysics software.

### 3.1. Boundary Condition Setting

Because the pore diameter in the porous oil-containing cage is at micron and sub-micron levels, the flow speed of the lubricating oil is so slow that the Reynolds number is very small. Therefore, “peristaltic flow” is selected as the physics interface. The calculation equations include the Stokes equation (Navier–Stokes) for the conservation of momentum and the Continuous equation for the conservation of mass. It is noted that the inertial term of the Navier–Stokes equation is ignored. Due to the huge change of pressure in the porous medium, the fluid is set to weakly compressible flow, and centrifugal force is added in the y-direction as in Formula (4), where *ω = r × π/30* (r(r/min) is the rotation speed.
(4)$$F_{ca}=\rho \left(y+0.028\left[m\right]\right)\omega^{2}\;\left[N/m^{2}\right]$$
where *ρ* is the density of lubricating oil determined by the material properties, 0.028 m is the radius of cage determined by the 7008C bearing size. Friction exists among the porous oil-containing cage, the balls, inner and outer rings, and the heat generated by friction causes the temperature of cage to rise. Therefore, based on the “peristaltic flow” interface, a “fluid heat transfer” interface is added to simulate the flow of lubricating oil. The equation is a generalized heat transfer equation, which is also called the energy conservation equation.

Finally, “Multiphysics coupling” interface is added, including flow coupling and temperature coupling. Flow coupling applies *u* and *p* of flow equation to the heat transfer equation, and temperature coupling applies *T* to the flow equation, which takes the effect of temperature on material properties (fluid viscosity, density, etc.) into account. In this two-way coupling mode, the flow state of lubricating oil in the porous oil-containing cage is simulated under the comprehensive influence of centrifugal force and heat effect. The boundary conditions are set as shown in Figure 4, where the upstream fluid temperature at the inlet boundary is 5 °C higher than the ambient temperature to represent the temperature rise of the porous oil-containing cage due to friction heat.

### 3.2. Analysis of Simulation Result

The transient solver is selected to analyze the unsteady flow field, including pressure field and temperature field. Additionally, the time step and the calculation time are set as 0.025 min, 3 min, respectively. The flow velocity and pressure distribution are calculated by the Multiphysics coupling method, and the results are shown in Figure 5. From top to bottom, the porous models with effective porosity of 0.1534, 0.1913, 0.2345, and 0.2848 are listed in order.

The changes of flow velocity and pressure drop inside the micropores of porous media in Figure 4 indicate that the flow law inside the porous media generally is in accordance with the law of porous seepage, that is, the flow velocity increases where the pressure difference is large. In addition, the difference between micropore distribution and geometric size leads to significant changes in local flow velocity and pressure drop, as described in [20]. Thus, macroscopic tribology experiments and porosity parameters are difficult to use to accurately reflect the internal flow mechanism of porous media. The modeling method in this paper provides a powerful analytical tool for the research of lubrication mechanism of porous oil-containing cages.

### 3.3. Mesh Independence Verification

To ensure the calculation results, mesh independence verification was carried out in this paper. The mesh refinement was done on the basis of the mesh model (the number of mesh elements was 148,089, porosity is 0.2848), and the number of mesh elements after the refinement was 207,467 and 311,329, respectively. Then, the simulation analysis of the model after the mesh refinement was carried out and the distribution of pressure is shown in Figure 6. It can be seen that the refinement of the mesh has little effect on the pressure distribution of the model, thus ensuring the accuracy of the simulation results.

## 4. Resistance Model of Micro-Scale Porous Media

The current theoretical models of flow resistance in the porous media feature a few problems, such as some empirical constants without physical meaning, limitation for micron/submicron structures, and the unspecified relationship between resistance and pore shape. Based on the established two-dimensional models of porous oil-containing cages with different porous parameters, this chapter develops a micro-scale resistance model for fluid flow in a micro-scale pores model of porous oil-containing cages based on the Hagen–Poiseuille equation.

### 4.1. Theoretical Derivation

Poiseuille [21] experimented with animal blood in a round tube (diameter 30–140 μm), and it was found that the volume flow *Q* per unit time was proportional to the pressure difference *(P*_1_ – *P*_2_*)* and the fourth power of tube radius *R*, and was inversely proportional to the length *L* of the tube. The well-known Hagen–Poiseuille equation was proposed as Equation (5).
(5)$$Q=\frac{\pi \left(P_{1}-P_{2}\right)R^{4}}{8\mu L}$$

This equation is applicable for the flow law in a micron-scale channel. In order to further research the lubrication mechanism of a porous oil-containing cage, the complex pore structure inside the porous medium is first simplified into a superposition model of multiple micro-scale circular tubes, and the flow resistance inside the porous medium is regarded as the sum of the resistance caused by each tube. Therefore, based on the Hagen–Poiseuille equation, the average flow velocity in the porous structure is derived from the flow rate and the cross-sectional area of the pipe, as shown in Equation (6).
(6)$$v_{ave}=\frac{q}{A}=\frac{\pi r^{4}}{8\mu L}\Delta P$$
where *A* is the cross-sectional area of the circular tube, *ΔP* represents the pressure difference between the outlet and the inlet, *μ* is the dynamic viscosity of the fluid (a fluid property related to temperature), *r* represents the radius of a uniform circular tube and *L* represents the length of the circular tube.

Because of the intricate distribution of micro-scale pores in the porous oil-containing cage and the uneven cross-section of tubes, the “average hydraulic radius” *R_h_* is used to replace the radius *r* of the circular tube. The hydraulic radius refers to the ratio of the overflow area of a certain water transmission section to the side length of water pipeline, as Equation (7).
(7)$$R_{h}=\frac{S_{all}}{Z_{all}}=\frac{\pi r^{2}}{2\pi r}=\frac{r}{2}$$
where *S_all_* is the total cross-sectional area of fluid flow and *Z_all_* is the total wetting circumference.

As the pore structures in the porous medium are curve channels instead of straight lines, the tortuosity parameter *τ = L_t_ /L* is defined to correct the length of tube. From Equation (7), Equation (6) can be converted into Equation (8).
(8)$$v_{ave}=\frac{q}{A}=\frac{\pi R_{h}^{2}}{2\mu L\tau }\Delta P$$

Further, according to the definition of hydraulic radius, Equation (9) can be derived.
(9)$$R_{h}=\frac{S_{all}}{Z_{all}}=\frac{S_{all}\times L_{t}}{Z_{all}\times L_{t}}=\frac{\frac{S_{all}\times L_{t}}{V_{t}}}{\frac{Z_{all}\times L_{t}}{V_{t}}}$$
where *L_t_* is the length of the channel, and *V_t_* is the overall volume of the porous medium, also known as the apparent volume.

The porosity of porous materials is divided into two types, namely, absolute porosity and effective porosity. Absolute porosity is the sum volume fraction of the connected pores and unconnected pores. From the perspective of fluid motion, only the interconnected pore space is meaningful, so the volume fraction composed of interconnected pores space is the effective porosity (such as *ε* = $\frac{S_{all}\times L_{t}}{V_{t}}$) in Equation (9). The porosity mentioned in this article is all the effective porosity, which is the ratio of the total volume *V_p_* of the connected pores to the overall volume *V_t_*, as in Formula (10):(10)$$\varepsilon =\frac{V_{p}}{V_{t}}$$

Next, *a = (Z_all_ × L_t_)/V_t_* is defined, and then Equations (9) and (10) are considered, from which Equation (11) can be obtained.
(11)$$R_{h}=\frac{\varepsilon }{a}$$
where *a* is the ratio of the total wetted area to the apparent volume of porous medium, which is a three-dimensional structural parameter in m^−1^. In order to keep the unit consistent without changing its physical meaning, *a* is converted into a two-dimensional structural parameter, which is the ratio of the total wetted perimeter to the apparent area of porous model, as shown in Equation (12).
(12)$$a=\frac{C_{p}}{s_{t}}$$
where *C_p_* is the total wetted circumference, which is the total boundary length of the porous model, and *S_t_* is the total area of the two-dimensional cross-sectional porous model. From Equations (11) and (12), *R_h_* is substituted into Equation (8), the micro-scale resistance model of porous media is derived as Equation (13).
(13)$$\frac{\Delta P}{L}=2{\left(\frac{C_{p}}{s_{t}}\right)}^{2}\frac{\mu \tau }{\varepsilon^{2}}v_{ave}$$

The flow velocity of the fluid through the micro-scale porous medium is very low, and it belongs to the range of peristaltic flow with a Reynolds number far smaller than 1. Therefore, the pressure drop generated by fluid unit through the pores that ignores inertial force loss is determined by the viscous drag loss, which is consistent with the actual theory. Besides, the pressure drop of the micro-scale resistance model has a linear relationship with the flow velocity, which is in accordance with Darcy’s law. It can be concluded from the above discussion that the micro-scale resistance model of porous media is reasonable.

### 4.2. Parameter Calculation

The micro-scale resistance model is a formula that includes parameters such as specific surface perimeter, effective porosity, tortuosity, fluid properties, fluid velocity and pressure drop. For the sake of further studying the lubrication mechanism in micro-scale porous media, the expression, composed of the parameters related to the porous structure and the fluid properties, is independently defined as a structural function *f(C_p_*, *S_t_*, *τ*, *μ*, *ε)* in Formula (14).
(14)$$f\left(C_{p},S_{t},\tau ,\mu ,\varepsilon \right)=2{\left(\frac{C_{p}}{s_{t}}\right)}^{2}\frac{\mu \tau }{\varepsilon^{2}}$$

Among them, *C_p_*, *S_t_* and *ε* are all determined by the geometric size and porous morphology of micro-scale pores model, *μ* is determined by the flow properties of the lubricating medium in the porous media, and *τ* is determined by the shape of the tubes in the porous model.

The way to obtain the tortuosity of porous media mainly includes extracting from 3D images, porosity derivation, and simulating fluid flow paths. In this paper, different methods of particle tracking and streamline acquisition are used to obtain the tortuosity of porous model.

#### 4.2.1. Particle Tracking

The particle tracking module of COMSOL calculates the movement of particles in the fluid media to obtain the movement trajectory of particles. Taking the porous model with an effective porosity of 0.1534 as an example, the inlet boundary is set to uniformly release 100 solid particles with a diameter of 0.0001 μm, and the outlet boundary is set to wall condition of freezing. The particles in the entire porous medium domain have an upward movement speed. Additionally, the wall conditions are mixed diffuse reflection and specular reflection.

The particles release at the entrance and gradually move toward the exit boundary. However, due to the tortuous complexity of the porous model, the particles repeatedly collide with the wall during the movement and rebound. Few particles could overflow the porous medium domain. Therefore, not all particles have reached the exit boundary. In order to obtain the porous parameters, the simplest trajectory of three particles is selected from the particles reaching the exit boundary, as shown in Figure 7.

The gray part in Figure 7 is the porous structure, and the thick blue line is the particle trajectory. Although the obtained particle trajectory cannot characterize the morphological changes of the porous structure in some details, the general trend is consistent with the porous medium model, which can be approximated as the basic data of the porous structure. However, due to the particles repeatedly bouncing against the wall while moving in the pores, the calculated particle trajectory is clearly larger than the actual length of the porous channel, and it needs to be corrected before it is used.

#### 4.2.2. Streamline Extraction Method

In order to accurately extract the porous channel length of micro-scale pores model, the streamline data are extracted based on the fluid flow path in the pore structure. As an example, the porous model with an effective porosity of 0.1534 is shown in Figure 8, which is a streamlined diagram of fluid flow in the porous model.

Figure 9 is the channel path diagram extracted from Figure 8, where the different colors stand for the segmented channels. Compared with the particle tracking method, the extraction method of streamline data is more accurate, and the error between the extracted channels’ data and the channels’ shape of the pores structure is smaller. Therefore, this method is used in this paper to obtain the tortuosity parameters of micro-scale pore model.

Due to the large number of segmented channels, Figure 9 only shows the extraction diagram of four connected channels. Based on the channel data stored in the form of x- and y-coordinate data, a program to calculate the actual length and visible length of the channel is written. The algorithm principle is shown in Figure 10.

Because there are enough data points for the segmented channels, the Pythagorean theorem is used to calculate the straight-line distance between the data points, which is regarded as an approximation length of the curved channel. Finally, the mean tortuosity of the segmented channels is applied to the structure formula in this paper.

The apparent area *S_t_* of porous model in this paper is the modeling space of 10,000 μm^2^. When the temperature is at 25 °C, the dynamic viscosity of the selected lubricating oil is 0.54 Pa∙s. Other porous parameters are listed and the structure function value *f* is calculated according to Formula (14), as shown in Table 1. The *f* is the structure function which is related to the geometric size of micro-scale porous model, the flow properties of the lubricating medium in the porous model, and morphological characteristics of pore structure. It can be seen that there are no empirical coefficients without physical significance in the above equations, which is of great significance for the flow mechanism of micro-scale porous media.

## 5. Model Verification

When the simulation time of fluid heat transfer in the micro-scale porous structure reaches 3 min, the velocity field, pressure field and temperature field of the porous model are stabilized. The average surface velocity of the porous model and the pressure drop between the outlet and the inlet are extracted. The ratio of the pressure drop to the speed is calculated, which is compared with the results calculated for *f* function in Section 3.

It can be seen form Table 2 that the difference between the simulation result of multiphysics coupling and the calculation result of micro-scale resistance model in pore structure derived in this paper is less than 6.5%, which is within the allowable range. It is indicated that the modeling method, simulation method and micro-scale resistance model of micro-scale pore model in this paper are reasonable, which provides a new direction for the research on the lubrication mechanism of porous oil-containing cages.

## 6. Conclusions

Based on the preparation principle of the porous oil-containing cage, the two-dimensional model of porous cage with high reduction degree was established by using random seeds theory. The Multiphysics coupling simulation analysis of the lubricating oil in the porous cage was carried out under the influence of centrifugal force and thermal effect. In order to further research the flow mechanism in the porous structure, based on the Hagen–Poiseuille equation, a micro-scale resistance model for fluid flow in the porous oil-containing cage was derived. The specific conclusions are as follows:
According to the preparation principle of porous cage, a digital modeling method of micron and submicron porous structure model is proposed based on random seed theory.Based on the above models, the influence of centrifugal force and thermal effects on the internal flow characteristics of the porous media is analyzed, and the flow law of the lubricating oil in the porous cage is obtained by multiphysics coupling analysis.A micro-scale resistance model without any empirical coefficients is proposed, which includes new structural parameters representing the geometric size of the porous model, the flow properties of the lubricating medium in the pores and the shape of the micro-scale pore structure.

## Figures and Tables

**Figure 1 materials-14-03896-f001:**
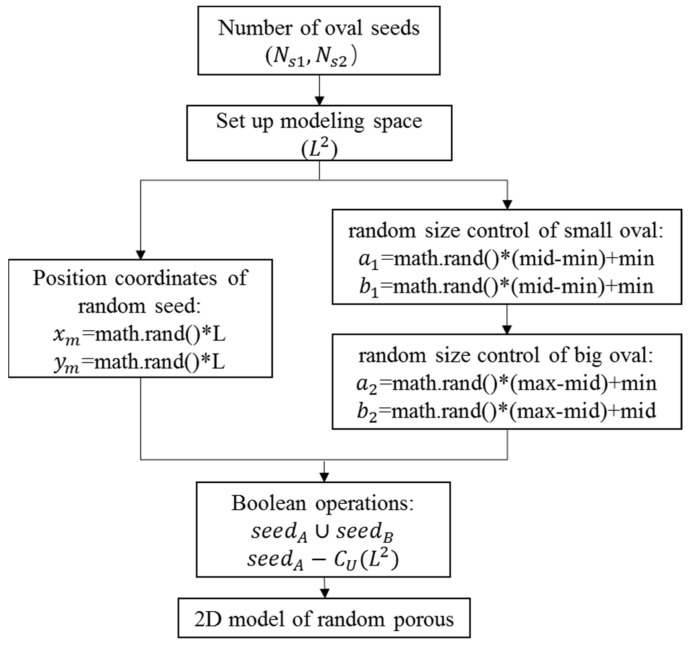
Algorithm flowchart.

**Figure 2 materials-14-03896-f002:**
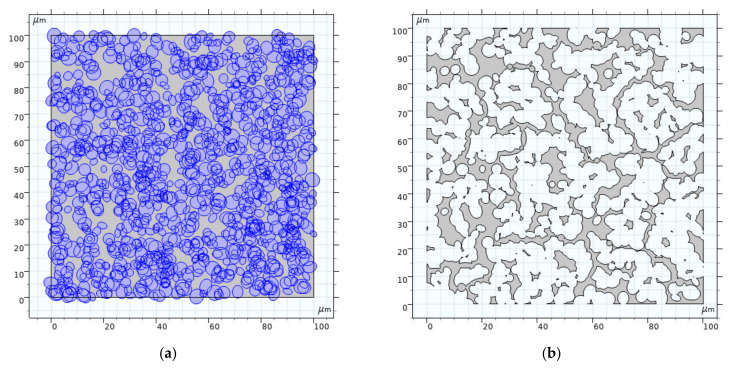
Porous structure model: (**a**) Random seeds accumulation; (**b**) Porous structure.

**Figure 3 materials-14-03896-f003:**
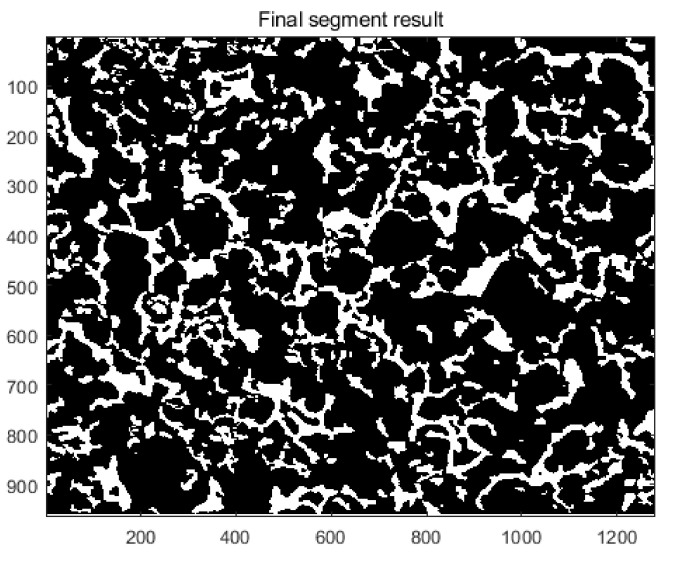
The binary graph of porous material.

**Figure 4 materials-14-03896-f004:**
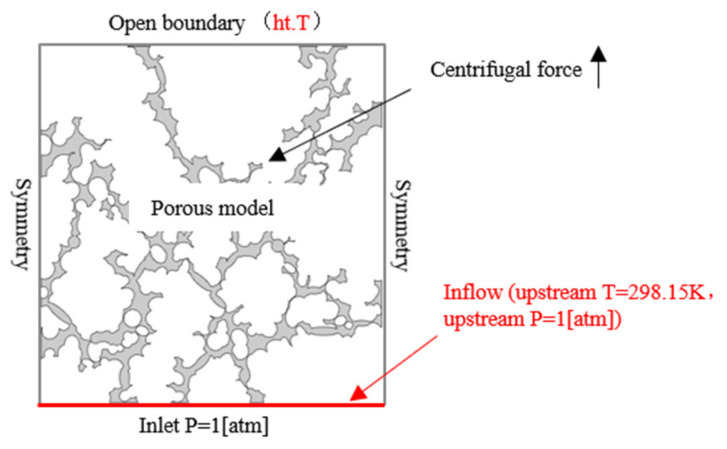
Boundary conditions setting of simulation.

**Figure 5 materials-14-03896-f005:**
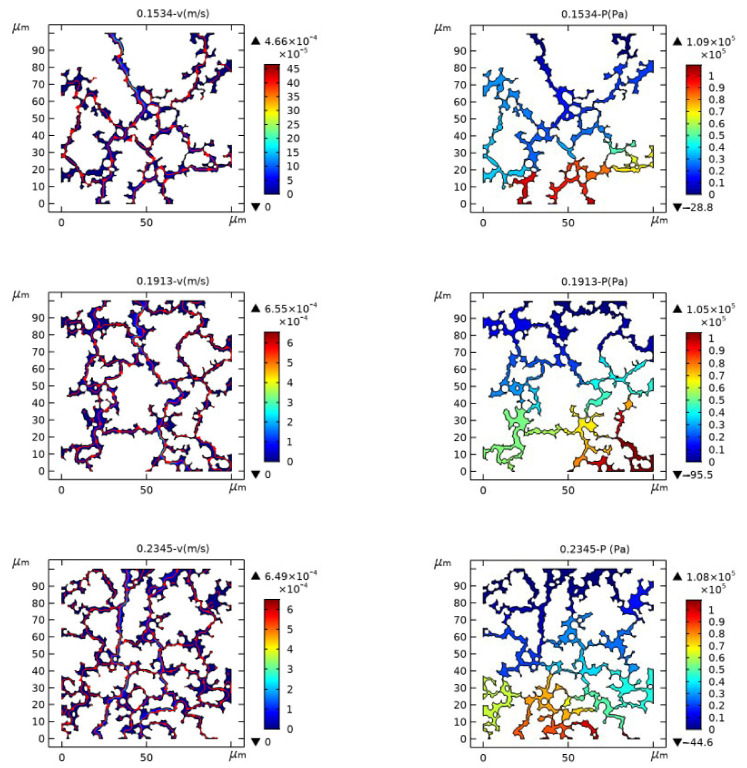
Distribution of flow speed (**left**) and pressure (**right**) of the porous model.

**Figure 6 materials-14-03896-f006:**
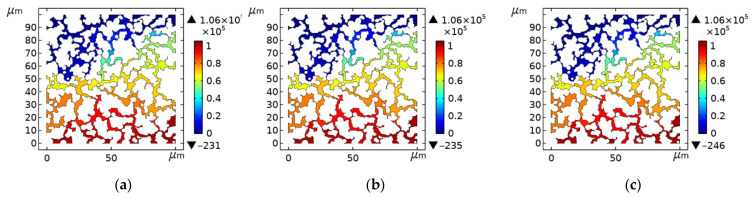
Distribution of pressure for models with different mesh densities: (**a**) Number of mesh elements: 148,089; (**b**) Number of mesh elements: 207,467; (**c**) Number of mesh elements: 311,329.

**Figure 7 materials-14-03896-f007:**
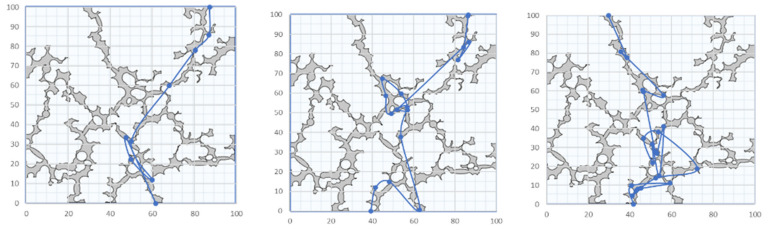
The simplest trajectory extraction diagram of three particles.

**Figure 8 materials-14-03896-f008:**
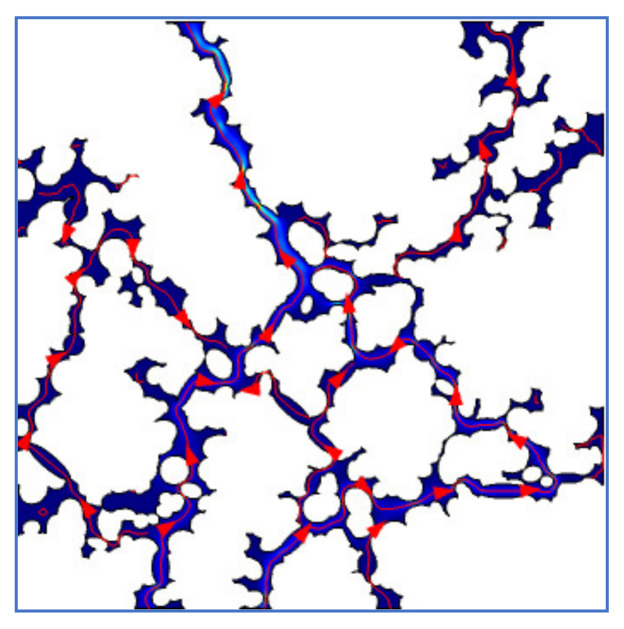
Streamline diagram.

**Figure 9 materials-14-03896-f009:**
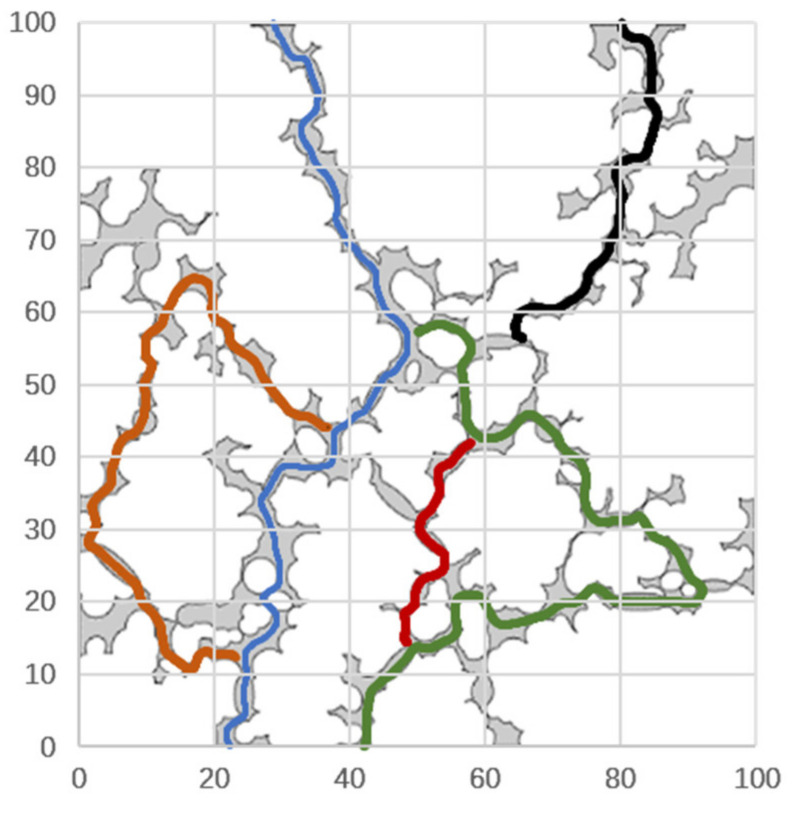
Extraction diagram of porous path.

**Figure 10 materials-14-03896-f010:**
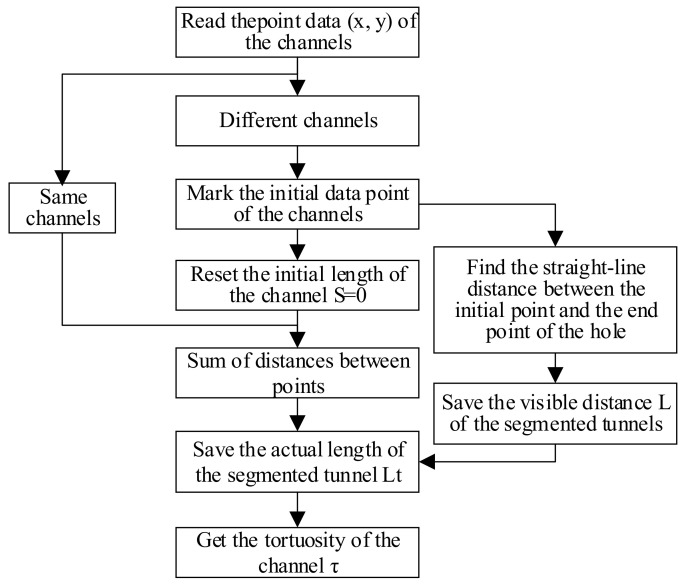
Schematic diagram of the algorithm for obtaining tortuosity.

**Table 1 materials-14-03896-t001:** The parameters of the porous structure.

Serial Number of Porous Models	Effective Porosity *ε*	Mean Tortuosity *τ*	Channel Perimeter *C* (μm)	*f* (×10^13^) (Pa∙s∙m^−2^)
1	0.1534	26.618	2039.1	5.080
2	0.1913	23.691	2376.8	3.950
3	0.2345	22.526	2890.0	3.695
4	0.2848	20.904	3622.5	3.652

**Table 2 materials-14-03896-t002:** The parameters of the porous structure.

Serial Number of Porous Models	Effective Porosity *ε*	$\frac{\Delta P}{L}\times \frac{1}{v_{ave}}(\text{×}10^{13})$ (Pa∙s∙m^−2^)	*f* (×10^13^) (Pa∙s∙m^−2^)	Error
1	0.1534	5.12	5.080	0.79%
2	0.1913	3.71	3.950	6.46%
3	0.2345	3.77	3.695	1.99%
4	0.2848	3.72	3.652	1.55%

## Data Availability

The data presented in this study are available on request from the corresponding author. The data are not publicly available due to the fact that they are also part of an ongoing study.

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
