# Peer review of "Flow Mechanism Characterization of Porous Oil-Containing Material Base on Micro-Scale Pore Modeling"

_materials, 2021, doi:10.3390/ma14143896_

Round 1
Reviewer 1 Report
In this manuscript, a new theoretical model for oil flow in porous lubricating medium for bearings is presented. Authors conduct a thorough algorithmic analysis of lubricating porous cage materials based on CT scan at various conditions of temperature and pressure. Effective porosity and area, pore tortuosity are all investigated parameters. The model and the results are very interesting as cage-containing oil are important lubricating medium for bearings in aerospace and similar extreme applications involving high pressure and temperature gradients. The model represents an improvement over previously description semi-empirical methods. This work should appeal to a broad audience of engineers and scientists in the fields of tribology and ceramic composites as lubricating materials. I have only minor recommendations, which are all language related. There are several sentences that need to be rewritten. Authors should make better text references to their figures, not all are called in the text. Authors must also more clearly describe how the porous cage is used for lubrication; as is, the reader will have difficulties understanding the purpose of the work. The title is also a bit confusing. Some of the mentioned sentences are marked below. Once these issues are addressed, the manuscript is recommended for publication.
Example of confusing sentences:
The main reason is that, the porous materials have structural characteristics such as randomness, complexity, micron/submicron scale, great difficulty in its structural modeling.
Literature [18] used X-ray CT technology to scan the internal pores structure of porous oil-containing cage, and found that the small pores in the porous oil-containing cage accounted for majority, and the dense distribution of small pores was the main reason for the connectivity of pores in porous oil-containing cage.
Because the return value of Math. Random () function is a random double-precision floating-point number which is between 0.0 and 1.0. The size of generated seeds fluctuates among the specified sizes(?????∈[???,???]).
Besides, the pressure pressure-drop of the micro-scale resistance (pressure is written twice)
However, due to the particles repeatedly bounced against the wall (it should read "bouncing")
Reviewer 2 Report
The paper is about flow analysis in the pore of a porous medium in the micro-scale level. I think that the paper is valuable and can be published after a small revision.
We had studies on the pore scale analysis of fluid flow in a porous medium. There were published in the following papers,
DOI: 10.1615/JPorMedia.2019028887
If the authors look at those papers, they will find that there are many methods that authors can use to be sure of their computational results such as
a) how many meshes exist inside the pore? Or in a cross section of a pore?
b) is there any difference between the inlet and outlet flow rate?
c) draw the pressure drop in flow direction, is it linear?
d) Mesh refinement
etc.
My only question is that how they can be sure of their computational results? Did they do any study to be sure of their results?
Author Response
请参阅附件。

Round 2
Reviewer 2 Report
I think it is nice study and it can be published.